# Molecule-Edit Templates for Efficient and Accurate Retrosynthesis Prediction

**Mikołaj Sacha**[1,2]    **Michał Sadowski**[1]    **Piotr Kozakowski**[1]
Ruard van Workum[1]    Stanisław Jastrzębski[1]

[1]Molecule.one, Poland
[2]Doctoral School of Exact and Life Sciences, Jagiellonian University
E-mail: {`m.sacha, s.jastrzebski`}`@molecule.one`

## Abstract

Retrosynthesis involves determining a sequence of reactions to synthesize complex molecules from simpler precursors. As this poses a challenge in organic chemistry, machine learning has offered solutions, particularly for predicting possible reaction substrates for a given target molecule. These solutions mainly fall into template-based and template-free categories. The former is efficient but relies on a vast set of predefined reaction patterns, while the latter, though more flexible, can be computationally intensive and less interpretable. To address these issues, we introduce METRO (Molecule-Edit Templates for RetrOsynthesis), a machine-learning model that predicts reactions using minimal templates - simplified reaction patterns capturing only essential molecular changes - reducing computational overhead and achieving state-of-the-art results on standard benchmarks.

## 1 Introduction

Retrosynthesis involves the strategic breakdown of complex molecules into simpler precursors, paving the way for the synthesis of novel molecules. Recently, there has been a development of AI-based methods for retrosynthesis, which allow learning reaction rules from the data of historically performed reactions. A central component of such systems is a model for *single-step* retrosynthesis that predicts what reactions could lead to a considered target molecule.

Two dominant methodologies are used for single-step retrosynthesis. *Template-based* methods use a set of translation rules that represent the possible chemical transformations. Although these methods are characterized by speed and interpretability, they may require an extensive set of templates to cover a large space of chemical reactions, which limits their generalization capacity. Conversely, template-free approaches can produce arbitrary reactions without such constraints but are often computationally demanding, largely due to their dependency on autoregressive decoding [1, 2, 3, 4]. This dichotomy highlights a research gap: there is a clear need for a method that leverages the structure of templates but also possesses the capability to generalize across diverse reactions. Such an approach would offer a promising balance between efficiency and accuracy in retrosynthetic predictions.

The current template-based strategies typically rely on templates that include the neighborhood of the reaction site. We note, however, that contemporary deep learning architectures are adept at processing the molecule's graph structure to evaluate the applicability of a given template at a given reaction site, rendering the inclusion of neighborhood information in templates redundant. Leveraging this insight, we developed METRO (Molecule-Edit Templates for RetrOsynthesis) - a model predicting minimal templates that include only the essential modifications to be made to a molecule, eliminating superfluous context. This enables METRO to use a smaller and more general set of templates, which

NeurIPS 2023 AI for Science Workshop.

makes the model more efficient and allows it to better cover the reaction space, as well as achieve state-of-the-art accuracy on standard retrosynthesis benchmarks. Moreover, we canonicalize the order of actions in the molecule-edit templates using a method based on the Weisfeiler-Lehman algorithm [5], which further improves the efficiency of our approach.

Our contributions can be summarized as follows:

1. We introduce METRO, a single-step retrosynthesis model that uses novel minimal molecule-edit templates and achieves state-of-the-art performance on standard retrosynthesis benchmarks. METRO uses fewer templates than prior works while maintaining the same or higher coverage of the reaction test set, which facilitates training on large-scale datasets.

2. We derive a template canonicalization method for molecule-edit templates based on the Weisfeiler-Lehman algorithm, which removes redundant templates and allows for more efficient training.

## 2    Related work

While the ultimate goal in retrosynthesis planning is to devise accurate multi-step methods that can predict the whole synthesis pathway, separating the problem of single-step retrosynthesis prediction can serve both as a useful benchmark and a step towards a robust multi-step system. Some research has shown that this separation yields more accurate results compared to directly modeling the entire multi-step process [6, 7]. Additionally, evaluating single-step models is more straightforward, usually facilitated by computing the top-k accuracy on datasets of chemical reactions. In this section, we summarize the prior work in single-step retrosynthesis prediction.

The early works relied on generating reactions predominantly using manually crafted rules. While effective, the meticulous process of crafting these rules was labor-intensive and required significant expertise [8, 9]. Another distinct branch of methods grounded itself in physical chemistry calculations [10]. Though precise, these methods were often computationally intensive, limiting their applicability. In response to these challenges, statistical approaches appeared that leverage vast datasets to make informed reaction predictions. These methods can be broadly categorized into template-based and template-free approaches.

**Template-based** approaches utilize reaction templates, predefined patterns representing chemical transformations. These can either be derived from a database of known reactions or manually specified. For example, Coley et al. [11] proposed a method to select reaction templates for the target molecule using a similarity metric. Segler et al. [12] and Baylon et al. [13] introduced neural networks as multiclass classifiers for selecting templates. GLN [14] models compounds as graphs and processes them with a graph neural network that learns the joint distribution of templates and targets. LocalRetro [15] also uses a graph neural network but simplifies the template by removing its neighborhood information. Another approach is RetroKNN [16], which boosts the performance of template-based systems by non-parametric retrieval, using a k-nearest-neighbor (KNN) search on reaction templates during inference.

**Template-free** methods formulate retrosynthesis prediction without the explicit usage of reaction templates. Some of these methods represent improvesreaction prediction as a SMILES to SMILES translation problem and adopt models from natural language processing to solve it. One example is Molecular Transformer [1] which applies the Transformer architecture [17] to this task. Tetko et al.[2] showed that SMILES augmentation boosts the accuracy of the model. Zhong et al. [3] used *root-aligned* SMILES representation that improves the learning ability of Transformer. RetroPrime [18] divides retrosynthesis prediction into two stages - decomposing a molecule into synthons, and generating reactants by attaching the leaving groups. Shi et al. [19] introduced a graph-to-graph (G2G) translation method that uses a graph neural network to split the target molecule into synthons and then translate them to the final reactants. MEGAN [4] represents a reaction as a sequence of actions that modify the input graph until the desired output is reached.

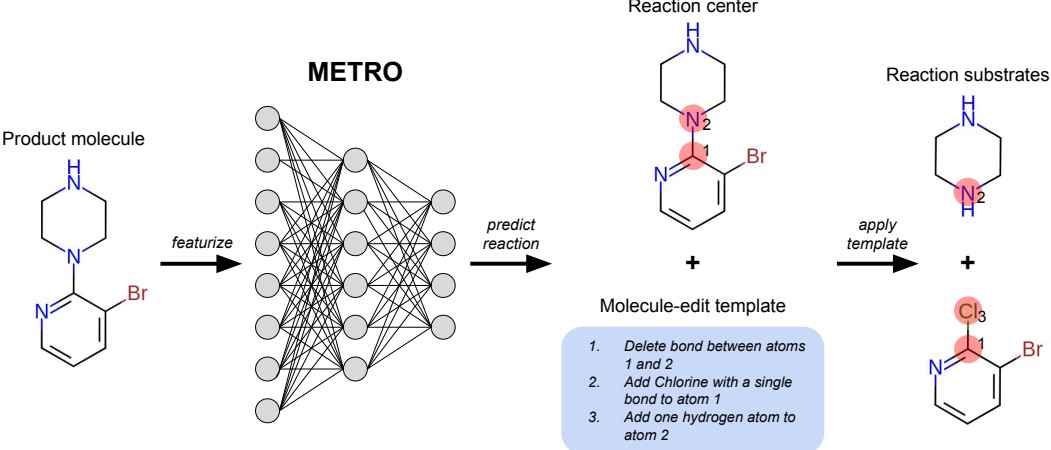

Figure 1: METRO model predicts the reaction center and the template to apply for a given product molecule. During inference, this information is used to apply the template which transforms the input molecule into the reaction substrates.

## 3 Method

### 3.1 Model architecture

To model single-step retrosynthesis, we build upon the neural network architecture presented in LocalRetro [15]. We show a schematic illustration of model inference in fig. 1. Similarly to LocalRetro, our model predicts *atom templates* or *bond templates*, which are applied to single atoms or ordered pairs of bonded atoms, respectively. The details about template definition and application are described in section 3.2. The input compound is represented as a graph, with nodes representing atoms and edges representing bonds. This graph is processed by a Message Passing Neural Network (MPNN) [20] to give a hidden representation $H_A \in \mathbb{R}^{|A| \times F}$ where $|A|$ is the number of atoms in the input compound and $F$ is the hidden dimension size. Next, we use the *global reactivity attention* (GRA) [15] mechanism to account for non-local relations in the input molecule. In our experiments, we found that, contrary to [15], it is beneficial to use GRA on the hidden features of atoms, not concatenated hidden features of atoms and bonds, as it achieves similar accuracy with faster inference. Namely, our GRA module computes

$$\mathbb{R}^{|A| \times F} \ni H_{GRA} = GRA(H_A), \tag{1}$$

where $GRA(x)$ consists of a single self-attention layer. Next, we concatenate features of neighboring atoms to obtain features of bonds. For each ordered bond between $i$-th and $j$-th we define its features as

$$\mathbb{R}^{2F} \ni b = H_{GRA}^i | H_{GRA}^j, \tag{2}$$

where $|$ indicates vector concatenation. Note that this gives us two different sets of features for each bond, depending on the order of concatenation of atom features ($ij$ or $ji$). We obtain a matrix $B_{GRA} \in \mathbb{R}^{2|B| \times 2F}$ containing features of bonds after GRA, where $|B|$ is the number of bonds in the input molecule. To get the final logits for templates per atoms and bonds, we pass $H_{GRA}$ and $B_{GRA}$ through two feed-forward layers

$$\mathbb{R}^{|A| \times F_A} \ni H'_{GRA} = \sigma(f(H_{GRA}))$$
$$\mathbb{R}^{|A| \times |T_A|} \ni H_{logits} = g(H_{GRA})$$
$$\mathbb{R}^{2|B| \times F_B} \ni B'_{GRA} = \sigma(h(B_{GRA}))$$
$$\mathbb{R}^{2|B| \times |T_B|} \ni B_{logits} = k(B_{GRA}),$$

where $\sigma$ indicates the ReLU activation function, $f$, $g$, $h$ and $k$ are standard feed-forward neural network layers, and $|T_A|$ and $|T_B|$ are the numbers of possible *atom templates* and *bond templates*, respectively. Finally, we flatten and concatenate $H_{logits}$ and $B_{logits}$ to get a vector of probabilities of templates applied to each relevant place in the input molecule

$$\mathbb{R}^{|A||T_A|+2|B||T_B|} \ni T_{pred} = Softmax(flatten(H_{logits})|flatten(B_{logits})) \tag{3}$$

## 3.2   Molecule-edit templates

In this section, we describe *molecule-edit templates* that we use to represent and predict chemical reactions in retrosynthetic direction. Similarly to other works, we do not consider additional reagents or procedural details and focus only on predicting the main substrates from a single reaction product. Consider a reaction $R$, defined as a pair of $(P, S)$, where $P = (V_P, \mathcal{E}_P)$ is a graph that represents the main reaction product, and $S = (V_S, \mathcal{E}_S)$ is a graph that represents the set of main substrates of the reaction, with $V$ and $\mathcal{E}$ denoting the sets of nodes and edges in a graph. We define a *Molecule-Edit Template* as a pair $T = (N, E)$ where $N \in \mathbb{N}$ is the number of product actions modified by the template and $E$ as the list of actions that need to be performed on the molecule $P$ to obtain the set of molecules $S$. In the following paragraphs, we describe how these actions are defined and how the molecule-edit templates are extracted and applied.

**Molecule edits.**    We define three types of *molecule edits*:

1. *AddAtom* adds a new atom to the input graph, bonded with a selected atom that already exists in the graph

2. *EditAtom* edits the properties of a selected atom in the input graph.

3. *EditBond* edits the properties of a selected bond in the input graph

Actions *AddAtom* and *EditAtom* are predicted per atom, and action *EditBond* is predicted per a pair of atoms. The properties of each action include information about the added or edited atom, or added or edited bond. We show the exact properties of each action type in the Supplementary Material.

**Extracting a molecule-edit template from a reaction.**    To build a database of molecule-edit templates for the METRO model, we extract the reaction template from each of the reactions from the training set. In fig. 2, we show an example template extracted from a reaction in retrosynthetic direction. We also show the outline of the template extraction procedure in algorithm 1. The result of this algorithm consists of the encoded template, together with the set of atoms in the reaction product modified by the template. This information allows us to train the METRO model in a supervised manner to simultaneously predict the reaction center and the number of the template to be applied, similar to [15].

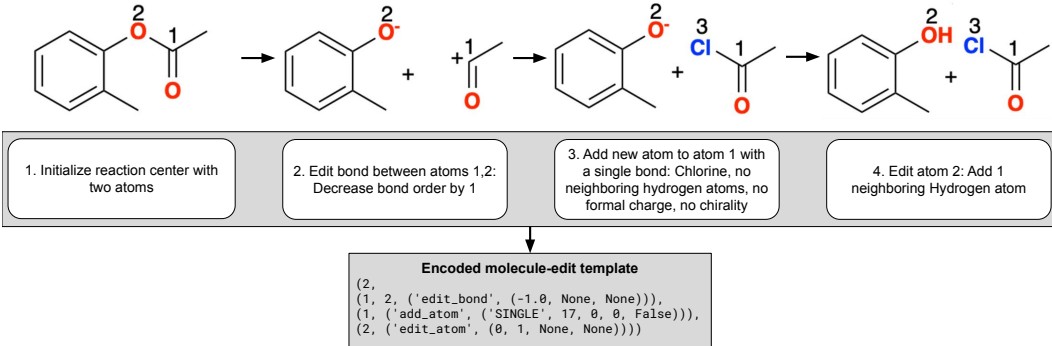

Figure 2: Example molecule-edit template extracted from an esterification reaction with acyl chloride, in retrosynthetic direction. The reaction is represented by a series of graph actions that modify the input molecule until the desired substrates are reached. These graph actions are encoded as a molecule-edit template in a machine-readable way.

---

**Algorithm 1** Extracting molecule-edit template from a reaction
---

1: **procedure** EXTRACTTEMPLATE($S, P$)            ▷ Where $R = (P, S)$ is the reaction
2:      $E \leftarrow [\,]$            ▷ Initialize empty list of edit actions E
3:      $C \leftarrow \emptyset$            ▷ Initialize empty set of modified product atoms
4:      $A_{\text{can}} \leftarrow CanonicalOrderOfAtoms(S, P)$            ▷ Get canonical order of atoms
5:      **while** $P \neq S$ **do**
6:          **for all** $a \in A_{\text{can}}$ **do**
7:              **if** an atom action $e$ exists on $a$ that modifies $P$ closer to $S$ **then**
8:                  $e \leftarrow$ ExtractAction($a$)            ▷ Extract the action as e
9:                  $P \leftarrow$ Apply($e, P$)            ▷ Apply e to P
10:                 Append $key(e)$ to $E$            ▷ Append the key of edit action $e$ to $E$
11:                 $C \leftarrow C \cup \{a\}$            ▷ Add the modified product atom to set C
12:              **end if**
13:              **for all** $a' \in A_{\text{can}}$ **do**
14:                 **if** a bond action $e$ exists on $(a, a')$ that modifies $P$ closer to $S$ **then**
15:                     $e \leftarrow$ ExtractAction($a, a'$)            ▷ Extract the action as e
16:                     $P \leftarrow$ Apply($e, P$)            ▷ Apply e to P
17:                    Append $key(e)$ to $E$            ▷ Append the key of edit action $e$ to $E$
18:                    $C \leftarrow C \cup \{a, a'\}$            ▷ Add the modified product atoms to set C
19:                 **end if**
20:              **end for**
21:          **end for**
22:      **end while**
23:      $T \leftarrow (|C|, E)$
24:      **return** $(T, C)$            ▷ Return the template T and modified product atoms C
25: **end procedure**

---

**Ensuring canonical molecule-edit templates.** In many cases, a sequence of molecule edits that defines a template can be applied in more than one order to achieve the same substrates. For example, two last actions from the template depicted in fig. 2 could be switched without changing the undergoing reaction. This phenomenon can lead to redundant templates and potentially lower expressiveness of the trained model. To overcome this, we introduce a method for finding a *canonical* order of atoms in the reaction. This canonical order is always used to select the next molecule-edit action when extracting a template from a reaction ($CanonicalOrderOfAtoms(S, P)$ in algorithm 1). This leads to the minimal possible number of templates covering the same reaction space without redundancy.

Given a reaction $R = (S, P)$, with $A$ being the set of all atoms from $P$ and $S$, we calculate three types of labels for the atoms $A$. Each label type is computed using the Weisfeiler-Lehman algorithm [5] for finding canonical labels of nodes in a graph. We define the *reaction center graph $C$* as the graph containing only the atoms that are modified during the reaction. Next, we compute canonical node

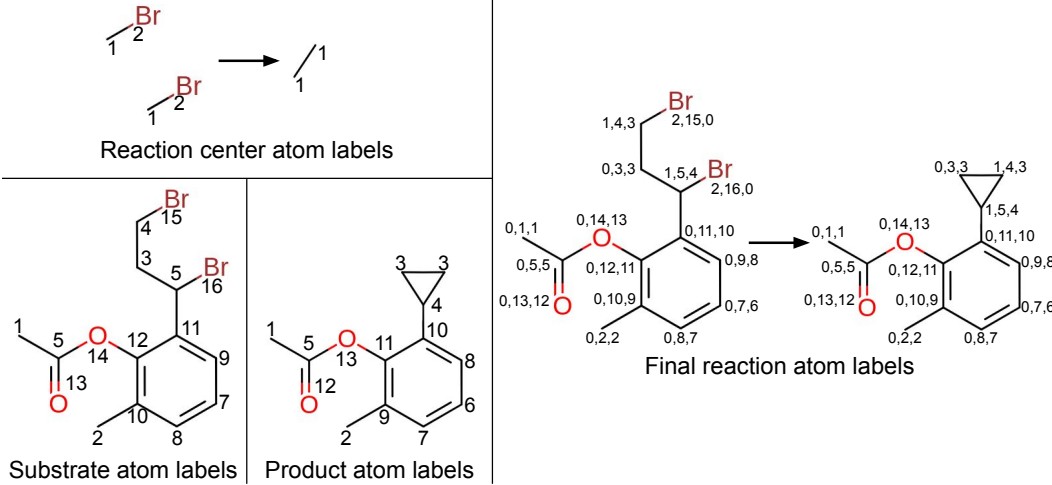

Figure 3: Example of computing canonical atom labels within a reaction using the Weisfeiler-Lehman algorithm. In this example, using the canonical labels computed only on the reaction center is not enough, as, for example, the Boron atoms are not distinguishable in such a case. Similarly, two of the three carbon atoms from the cyclopropyl group in the product are indistinguishable when considering only the reaction product. Both of these problems are combated when we take into consideration labels from all three of the graphs.

labels $L_C$, $L_S$ and $L_P$ using the Weisfeiler-Lehman algorithm separately on the reaction center graph, reaction substrates graph, and reaction product graph. To obtain the final atom labels, we merge the three acquired labels for each of the atoms in the reaction, acquiring node labels $L$, where for each $l \in L$, $l = (l_C, l_S, l_P)$. In cases where an atom did not appear in one of the graphs, we set its label in this graph to zero. The canonical order of atoms is obtained by sorting the final atom labels $L$ using lexicographic order. In fig. 3, we show an example of canonical labeling of reaction atoms using our method, and why using only the reaction center graph might not be enough to get truly canonical atom labels.

**Model inference.** Given an input reaction product $P$, we run the trained METRO model to acquire the matrix of probabilities of all the templates over all atoms and bonds $T_{pred}$. From this matrix, we retrieve the top $K$ outputs with the highest scores to acquire $K$ reaction predictions, sorted by their decreased predicted probability. For each prediction, we apply the predicted template on the predicted atom or predicted bond, depending on the template type. In the case of templates where there are more than two reaction center atoms in the product, we follow a similar protocol to [15], enumerating all possible sites in the product that match the first two atoms in the template reaction center definition. All such enumerated predictions have the final score $p/k$, where $p$ is the original score predicted by the model, and $k$ is the number of enumerated reactions.

## 4 Experiments

### 4.1 Experimental setup

We evaluate our method on three standard benchmarks for single-step retrosynthesis: USPTO-50k, USPTO-MIT, and USPTO-FULL. USPTO-50k contains 50 thousand reactions from 10 reaction types, while USPTO-MIT and USPTO-FULL are large-scale datasets containing about 480 thousand and 1 million reactions, respectively. We follow the standard train/valid/test splits for these datasets, introduced in [11, 21, 14]. For USPTO-50k, we also train the model for a variant with the reaction type provided in the input. In this case, we append the one-hot encoded reaction class as an additional node feature to all the input atoms. The model was trained on a single NVidia A100 GPU, with the training running from about 7 hours for USPTO-50k to about 10 days for USPTO-FULL.

Table 1: Top K test accuracy on retrosynthesis benchmarks. The best results are emboldened.

| Dataset & Method type | Method | Top K accuracy (%) | | | | | |
|---|---|---|---|---|---|---|---|
| | | 1 | 3 | 5 | 10 | 20 | 50 |
| *USPTO-50k* | | | | | | | |
| Template-based | GLN [14] | 52.5 | 69.0 | 75.6 | 83.7 | 89.0 | 92.4 |
| | METRO | **56.5** | **80.2** | **86.8** | **92.2** | **95.4** | **96.8** |
| Semi-template | RetroPrime [18] | 51.4 | 70.8 | 74.0 | 76.1 | — | — |
| Template-free | MEGAN [4] | 48.1 | 70.7 | 78.4 | 86.1 | 90.3 | 93.2 |
| | AT [2] | 53.5 | — | 81.0 | 85.7 | — | — |
| | Root-aligned [3] | 56.3 | 79.2 | 86.2 | 91.0 | 93.1 | 94.6 |
| *USPTO-50k ("stereo-aware" evaluation)* | | | | | | | |
| Template-based | LocalRetro [15] | 53.4 | 77.5 | 85.9 | 92.4 | — | 97.7 |
| | RetroKNN [16] | 57.2 | 78.9 | 86.4 | 92.7 | — | **98.1** |
| | METRO | **57.3** | **81.2** | **88.0** | **94.4** | **96.6** | **98.1** |
| *USPTO-50k reaction type given* | | | | | | | |
| Template-based | GLN [14] | 64.2 | 79.1 | 85.2 | 90.0 | 92.3 | 93.2 |
| | METRO | **67.9** | **88.7** | **93.0** | **95.9** | **97.1** | **97.5** |
| Template-free | MEGAN [4] | 60.7 | 82.0 | 87.5 | 91.6 | 93.9 | 95.3 |
| *USPTO-50k reaction type given ("stereo-aware" evaluation)* | | | | | | | |
| Template-based | LocalRetro [15] | 63.9 | 86.8 | 92.4 | 96.3 | — | 97.9 |
| | RetroKNN [16] | 66.7 | 88.2 | 93.6 | 96.6 | — | 98.4 |
| | METRO | **69.1** | **90.0** | **94.3** | **97.1** | **98.3** | **98.8** |
| *USPTO-MIT* | | | | | | | |
| Template-based | LocalRetro [15] | 54.1 | 73.7 | 79.4 | 84.4 | — | 90.4 |
| | RetroKNN [16] | **60.6** | 77.1 | 82.3 | **87.3** | — | **92.9** |
| | METRO | 59.4 | 76.2 | 81.0 | 85.6 | 88.6 | 91.2 |
| Template-free | Root-aligned [3] | 60.3 | **78.2** | **83.2** | **87.3** | **89.7** | 91.6 |
| *USPTO-FULL* | | | | | | | |
| Template-based | GLN [14] | 39.3 | — | — | 63.7 | — | — |
| | METRO | 45.5 | 60.0 | 64.0 | 68.2 | 71.2 | 73.8 |
| Template-free | MEGAN [4] | 33.6 | - | - | 63.9 | - | 74.1 |
| | AT [2] | 46.2 | - | - | 73.3 | - | - |
| | Root-aligned [3] | **48.9** | **66.6** | **72.0** | **76.4** | **80.4** | **83.1** |

To improve the generalizability of the model, we use cosine annealing learning rate schedule with stochastic weight averaging. For USPTO-50k training, we use a cyclical learning rate schedule, with the final model weights acquired by taking the average over the weights after each of the cycles. For USPTO-MIT and USPTO-FULL we run a single long cosine learning rate cycle, with the final weights acquired by taking the average over the weights of the 20 checkpoints that achieve the best validation accuracy. We present the exact values of model hyperparameters and a plot illustrating the learning rate schedule in the Supplementary Material.

## 4.2 Results

**Top K accuracy on USPTO benchmarks.** In table 1, we compare the performance of METRO with selected baseline methods, including the current state-of-the-art approaches. Following previous works, we compute the Top K accuracy score on the USPTO-based benchmark datasets. METRO achieves the best accuracy out of all models on the USPTO-50k benchmark, both with and without the reaction type given as prior information to the model. Particularly, METRO surpasses the accuracy of the *Root-aligned SMILES* model [3], which is a transformer-based model with 20 times test time augmentation during inference. METRO also achieves competitive accuracy on USPTO-MIT and

Table 2: Test set coverage with different template types extracted from the training set.

| Method | USPTO-50k | | USPTO-MIT | | USPTO-FULL | |
| --- | --- | --- | --- | --- | --- | --- |
| | No of temp. | Coverage | No of temp. | Coverage | No of temp. | Coverage |
| GLN [14] | 11647 | 93% | 80167 | 91% | 259494 | 78% |
| LocalRetro [15] | 731 | 98% | 21081 | 97% | 184952 | 83% |
| METRO | 627 | 99% | 13438 | 97% | 100232 | 83% |

Table 3: The number of templates extracted from the USPTO-50k training set, depending on what template canonicalization method is used.

| Template canonicalization | Number of templates | Test set coverage |
| --- | --- | --- |
| Random | 1073 | 98.4% |
| RdKit canonical SMILES | 679 | 99.0% |
| Weisfeiler-Lehman | 627 | 99.1% |

USPTO-FULL datasets, surpassing all the template-based models apart from RetroKNN [16]. As [15, 16] use a modified evaluation protocol for USPTO-50k, we compare them separately as the *stereo-aware* evaluation.

**The efficiency of molecule-edit templates.** To compare the efficiency of different reaction template types on the benchmark datasets, we extract all the reaction templates from the training set and calculate the fraction of the test set that can be covered using these templates. In table 2, we show this comparison for METRO and two previous state-of-the-art template-based approaches, GLN [14] and LocalRetro [15]. By representing templates only as molecular edits we can reduce the number of templates by a significant margin while retaining high test set coverage. This difference is particularly noticeable on large-scale datasets, such as USPTO-FULL, where METRO can cover a similar fraction of the test set with about 46% fewer templates, compared to LocalRetro [15].

**The importance of the canonicality of templates.** The canonical order of actions introduced in section 3.2 aims to minimize the number of templates needed to cover the reaction space. In table 3, we show a comparison of the number of templates extracted from the USPTO-50k training set, depending on the method of ordering molecule atoms when finding molecule-edit actions. Using simply a random order of atoms for each reaction yields 1073 templates, while following the order of atoms in the canonical molecule SMILES from the RdKit library [22] yields 679 templates. When using our canonicalization algorithm, the number of templates decreases to 629, and they cover the larger proportion of the test set.

## 5 Conclusions

In this work, we present Molecule-Edit Templates for Retrosynthesis (METRO), a model for reaction prediction that uses efficient minimal templates. We showcase that METRO allows for state-of-the-art accuracy on standard benchmarks with a smaller number of templates compared to the previous approaches. Additionally, we introduce an algorithm for ensuring the canonicality of the templates using the Weisfeiler-Lehman method for labeling graph nodes, which ensures that there are no redundant templates in the training set. Possible future work can involve improving the accuracy of the model on large scale datasets, for example by merging edit actions that commonly occur together in reactions, or combining our method with a template-free approach.

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
