# OpenReview forum: "Molecule-edit templates for efficient and accurate retrosynthesis prediction"
_NeurIPS.cc/2023/Workshop/AI4Science — NeurIPS2023-AI4Science Poster_

### Official Review · Reviewer_g6Kh · 2023-10-16
**A new molecule-edit template-based method**

**Rating:** 6
**Confidence:** 3

**Review:**

In this paper, the authors present Molecule-Edit Templates for Retrosynthesis (METRO), a model for reaction
prediction that uses efficient minimal templates. They showcase that METRO allows for state-of-the-art accuracy on (4/5) standard benchmarks with a smaller number of templates compared to the previous approaches.

In general, I would suggest the authors to provide more analysis on benchmarks that METRO didn't achieve the best and to better understand the potential advantage and usecase of METRO, in general this paper is above the acceptance line.

---

### Official Review · Reviewer_cybe · 2023-10-25
**Interesting method for reaction prediction with promising experimental results**

**Rating:** 6
**Confidence:** 4

**Review:**

This paper proposes a method for single-step retrosynthesis called METRO. The method essentially decomposes reaction prediction into a series of small graph edits. A neural network "policy" is learned to apply these edits.

The paper is fairly clearly presented and the core idea makes sense. The empirical results look good (although I am aware that it is easy to make mistakes in evaluation, e.g. accidentally providing atom mapping labels, ignoring stereochemistry in evaluation: I did not have time to check for this).

The main disadvantage is that the distinction between METRO and other "semi-template" methods like MEGAN is unclear. Although I forget the details of MEGAN, at a high level it is doing the same thing: learning to apply a series of small single graph edits. The authors do not clearly state what distinguishes METRO from MEGAN. My sense is that it is just a series of "small choices", e.g. the exact architecture or defining the action space. In a future version of this work I suggest that the authors compare more with MEGAN: both explain differences, and, given the similarities between the methods, ideally do experiments to see what causes the difference in performance.

I am happy to accept this paper to a workshop so I will give a score of of 6, but I would probably give a lower score if it were a full conference paper.

---

### Meta-Review · Area_Chair_AsxA · 2023-10-26

**Recommendation:** Accept (Poster)
**Confidence:** 5

**Metareview:**

This paper exploits graph edits to capture the minimal template of a reaction for retrosynthesis prediction.

Good paper! Accept.